# A Pilot Cross-Sectional Study of Immunological and Microbiome Profiling Reveals Distinct Inflammatory Profiles for Smokers, Electronic Cigarette Users, and Never-Smokers

**DOI:** 10.3390/microorganisms11061405

**Published:** 2023-05-26

**Authors:** Peter G. Shields, Kevin L. Ying, Theodore M. Brasky, Jo L. Freudenheim, Zihai Li, Joseph P. McElroy, Sarah A. Reisinger, Min-Ae Song, Daniel Y. Weng, Mark D. Wewers, Noah B. Whiteman, Yiping Yang, Ewy A. Mathé

**Affiliations:** 1Comprehensive Cancer Center, The Ohio State University and James Cancer Hospital, Columbus, OH 43210, USA; kying88@gmail.com (K.L.Y.);; 2Department Internal Medicine, The Ohio State University College of Medicine, Columbus, OH 43205, USA; 3Molecular, Cellular and Developmental Biology Program, The Ohio State University, Columbus, OH 43210, USA; 4Department of Epidemiology and Environmental Health, University at Buffalo, Buffalo, NY 14261, USA; 5Department of Biomedical Informatics, College of Medicine, The Ohio State University, Columbus, OH 43210, USA; 6Division of Environmental Health Sciences, College of Public Health, The Ohio State University, Columbus, OH 43210, USA; 7Pulmonary and Critical Care Medicine, Davis Heart and Lung Research Institute, The Ohio State University, Columbus, OH 43210, USA; 8Division of Preclinical Innovation, National Center for Advancing Translational Sciences, National Institute of Health, Rockville, MD 20892, USA

**Keywords:** inflammation, RNA-sequencing, metatranscriptome, vaping, bronchoalveolar lavage

## Abstract

Smokers (SM) have increased lung immune cell counts and inflammatory gene expression compared to electronic cigarette (EC) users and never-smokers (NS). The objective of this study is to further assess associations for SM and EC lung microbiomes with immune cell subtypes and inflammatory gene expression in samples obtained by bronchoscopy and bronchoalveolar lavage (*n* = 28). RNASeq with the CIBERSORT computational algorithm were used to determine immune cell subtypes, along with inflammatory gene expression and microbiome metatranscriptomics. Macrophage subtypes revealed a two-fold increase in M0 (undifferentiated) macrophages for SM and EC users relative to NS, with a concordant decrease in M2 (anti-inflammatory) macrophages. There were 68, 19, and 1 significantly differentially expressed inflammatory genes (DEG) between SM/NS, SM/EC users, and EC users/NS, respectively. *CSF-1* and *GATA3* expression correlated positively and inversely with M0 and M2 macrophages, respectively. Correlation profiling for DEG showed distinct lung profiles for each participant group. There were three bacteria genera–DEG correlations and three bacteria genera–macrophage subtype correlations. In this pilot study, SM and EC use were associated with an increase in undifferentiated M0 macrophages, but SM differed from EC users and NS for inflammatory gene expression. The data support the hypothesis that SM and EC have toxic lung effects influencing inflammatory responses, but this may not be via changes in the microbiome.

## 1. Introduction

Smoking continues to be a major driver of a multitude of health conditions, including cancer [1]. One hallmark of cancer is inflammation, and its impact on disease-related pathways is well documented [2,3,4]. Smoking combustible cigarettes causes both pro-inflammatory and anti-inflammatory effects in the lung [3,5,6]. A critical component of inflammation and disease pathways is the recruitment of monocytes to the lung, which differentiate into many subtypes of macrophages, including undifferentiated naïve M0 subtypes, pro-inflammatory M1 subtypes, and anti-inflammatory M2 subtypes [7]. Smoking cigarettes downregulates pro-inflammatory M1 macrophage states and induces anti-inflammatory M2 macrophage states in humans [8,9,10].

Since being introduced to the US market in the mid-2000s [11], electronic cigarettes (EC) have been marketed as a safer alternative to smoking combustible cigarettes. Clinical trials provide some evidence that vaping may foster smoking cessation [12,13]. However, the long-term health effects of vaping remain unknown. EC heat and aerosolized e-liquids are composed of a mixture, including nicotine, flavors, vegetable glycerin, and propylene glycol. Heating e-liquids yields volatile organic compounds, which have induced inflammation in experimental and human studies [14,15,16,17,18,19,20,21,22]. Using bronchoscopy with bronchoalveolar lavage (BAL), we have previously shown that never-smokers (NS) using nicotine- and flavor-free EC over four weeks have a measurable increase in lung macrophages and inflammatory cytokines [14]. In a separate cross-sectional study of 72 participants (including the participants reported herein), we reported statistically significant higher levels of inflammation-related biomarkers among smokers (SM), compared to EC and NS, who were similar to each other [15]. Other cross-sectional studies have reported positive associations between EC use and increased lung inflammation [16], changes in gene expression of alveolar macrophages, and small airway epithelial cells [17], as well as significant increases in innate defense proteins, such as elastase and matrix metalloproteinase-9 [16,18].

The lung microbiome can shape the immune response [23], and although its role in lung health and the development of disease is poorly understood; there is emerging evidence that an altered lung microbiome can promote lung carcinogenesis through inflammation [24,25,26,27]. Some studies have not found significant differences in the microbiome diversity for SM and NS [28,29,30]. We have previously shown that there are differentially abundant bacteria among SM, EC, and nonsmokers (NS), although the bacterial diversity is no different [31]. Separately, we have shown that gene expression profiles are different in bronchial epithelial cells of SM compared to EC and NS, where EC and SM have similar profiles [15].

The objective of this study is to further assess associations for SM and EC lung microbiomes with immune cell subtypes and inflammatory gene expression in samples obtained by bronchoscopy and bronchoalveolar lavage. Herein, we now report associations by tobacco use for immune cell subtypes and inflammatory gene expression in bronchoalveolar lavage (BAL) samples from SM, EC, and NS.

## 2. Material and Methods

### 2.1. Study Participants and Exclusion Criteria

The details of this cross-sectional study have been previously reported [15]. Briefly, between 2015 and 2017, 28 healthy adult participants (ages 21–30 years) underwent lung bronchoscopy and BAL. “Healthy” was defined as not having an immune system disorder requiring medication, clinically diagnosed pulmonary disease under therapy, kidney or liver diseases, or other medical disorders that would increase the risk from bronchoscopy, body mass index > 40, or affect biomarker data. A subset of these subjects was chosen for additional analysis based on gender, age, and sample availability. NS were participants (*n* = 10) who had smoked fewer than 100 combustible cigarettes in their lifetime, as defined by the Centers for Disease Control [32] and had not smoked a cigarette or used EC for ≥1 year. EC users (*n* = 10; 8 of whom were former smokers) were daily users for ≥6 months and had not smoked a cigarette for ≥1 year. Smokers were required to smoke ≥10 cigarettes/day for at least six months and had not used EC within one year. Fewer women SM had sufficient sample available for analysis. This study was approved by the Ohio State University (OSU) Comprehensive Cancer Center Clinical Scientific Research Committee and the OSU Institutional Review Board.

Demographic and clinical characteristics of study participants are summarized in Table 1. The study workflow is summarized in Appendix A.

### 2.2. Biospecimen Collection

Participants underwent a bronchoscopy at the OSU Clinical Research Center after an orientation visit and consent. Saline (100 mL) was instilled and recovered from either the right middle lobe or lingula. The recovered BAL fluid was placed on ice and within 30 min spun at 10,000 g to produce a cell pellet. The BAL supernatant was removed, and the cell pellets were resuspended in 1 mL of QIAgen RNAprotect cell reagent. The samples were stored at −80 °C.

### 2.3. RNA Extraction and Sequencing

RNA was extracted from the BAL cell pellets using miRNeasy kit (Qiagen, Crawley, UK). RNA integrity was assessed using Bioanalyzer (Agilent, Santa Clara, CA, USA). Total RNA was then treated with TruSeq Stranded Total RNA Gold (Illumina, San Diego, CA, USA) to remove rRNA; cDNA was then synthesized from the remaining total RNA, and the library was generated with the Illumina TruSeq kit. RNA-sequencing (RNA-seq) was performed with the Illumina Hi-seq 4000 at 2 × 150 bps. Samples were sequenced to at least 35 million total reads per sample (microbial and human reads). Samples were then demultiplexed and read and 2 FASTQ files were produced for each participant and uploaded onto BaseSpace.

### 2.4. Human Transcriptome Analysis

FASTQC was applied to FASTQ files to assess quality of reads. Reads with average PHRED quality scores ≥ 30 were retained. Cutadapt version (v.) 1.14 [33] was applied to trim adapter sequences. The remaining reads were then aligned to the human genome (hg19) by TopHat [34] v.2.1.0, using the following parameters: *-p 8 --no-coverage-search --library-type fr-firststrand hg19.genome Read1.fastq.gz Read2.fastq.gz*. To calculate human gene counts, reads that mapped to the human genome (hg19) were contained in a BAM file. The BAM file was provided to subread v.1.6.2 [35] with the following parameters: *subread-1.6.2-source/bin/featureCounts -p -a gencode.v19.annotation.gtf -o Featurecount.txt Sample_File.bam*. The gene count for each subject was then compiled into an overall text file with 57,820 human genes, which was used in downstream human gene analysis.

### 2.5. Inflammatory Gene Expression

The aligned human reads ranged from 13.8 million to 41.3 million sequenced reads that mapped to 57,820 human genes (Appendix A). The genes were filtered using the *filterByExpr* function in edgeR package to remove lowly present genes. The default parameters and minimal read count of 10 produced a list of 20,517 genes [36]. The gene list was then normalized to library size using the *calcNormFactor* function in the edgeR package v.3.26.8 [37]. This was followed by log2-counts transformations, which were calculated using the *voom* function in the limma package v. 3.40.6 [38], followed by library generation batch correction using the *combat* function in the sva package v. 3.32.1 in R [39]. Quality (e.g., possible outliers, batch effects) of the metatranscriptome data was evaluated using PCA (Appendix A).

The gene ontology (GO) term for inflammatory response (GO:0006954) was selected as GO term of reference. GO:0006954 gene list was provided by BioMart GRCh37 and consisted of 620 genes. The 620 inflammatory response genes were compared to our list of 20,517 human genes, and a list of 420 inflammatory genes was produced and used in pairwise differential gene expression analysis. Comparison of gene expression by smoking groups before and after selection of inflammatory genes were analyzed by PCA (Appendix A).

### 2.6. Immune Cell Subtype Populations

CIBERSORT computational algorithm was used to identify 22 immune cell subtypes by deconvoluting RNA-seq gene expression data. The immune cell subtypes included B cells, T-cells, natural killer cells, macrophages, eosinophils, and neutrophils [40]. CIBERSORT has been used in studies on BAL [41,42,43], and it is considered one of the best bulk RNA deconvolution methods [44]. Raw gene counts were normalized as fragments per kilobase of transcript per million (FPKM) and processed in CIBERSORT using the LM22 gene matrix, as suggested. Results were returned as percentages of each immune cell type.

### 2.7. Statistical Analysis of Immune Cell Subtypes

Immune cell subtype composition as computed by CIBERSORT was tested for significance using Kruskal-Wallis test for overall three smoking group comparison with FDR correction [45]. Immune cells with FDR corrected *p* < 0.05 were considered significant. Immune cell subtypes that were significant after FDR correction were then tested for pairwise significance by post hoc Dunn test comparison. Pairwise smoking group comparisons *p* < 0.0167 (0.05/3 pairwise comparisons) were considered significant.

### 2.8. Microbiome Metatranscriptomics

Reads that were not mapped to the human genome were aligned to complete genomes of microbes downloaded from the NCBI microbial genome database (https://ftp.ncbi.nlm.nih.gov/genomes/refseq/; dated March 2018). Kraken v. 1 [46] was used for alignment with the following parameters: *kraken --preload –Kraken_Database --fastq-input --gzip-compressed --paired Microbiome_Read1.fastq.gz Microbiome_Read2.fastq.gz --output Microbial_alignment*. Resulting alignment files were then converted into a read count table: *kraken-report -Kraken_Database Microbial_alignment*. Aligned reads by Kraken were then fed into Bracken v.1 to align to bacterial species using the following parameters: *python est_abundance.py -i BAL_Samples_Full_Kraken_subanalysis -k KMER_DISTR.txt*. Bracken [47] was used to re-classify all non-species classifications from Kraken to species level classifications.

Batch effects of the metatranscriptome data were evaluated using principal component analysis (PCA). Additionally, based on the PCA, one outlier, a smoker, was removed from further analysis (Appendix A). Bacterial reads were reclassified from bacteria species to bacteria genus using the aggregate function in R, which merged and reallocated all bacteria species counts to their identified bacteria genera. In the remaining 27 BAL samples, prefiltered sequences ranged from 71,440 to 351,058 sequences per sample and were then mapped to 997 unique bacterial genera. These were then filtered on the following two criteria: (1) bacteria genera present in at least 2 study participants and (2) aligned reads of the genera were ≥0.005% total aligned microbial reads in a study participant, resulting in 509 bacteria genera. The 509 bacteria genera were normalized to library size using the *calcNormFactor* function in the edgeR package v.3.26.8 [37]. This was followed by log2-counts transformations calculated using the *voom* function in the limma package v. 3.40.6 [38]. An observed library generation batch effect was adjusted through the *combat* function in the sva package v.3.32.1 (Appendix A) [39]. After filtering, normalization and correcting for library generation batch, the mapped reads ranged from 71,249 to 350,109 and were mapped to 509 unique bacteria genera (Appendix A). The final step was to remove the bottom 10% of bacteria genera based on coefficient of variation (CV) to produce a list of 458 bacteria genera for IntLIM association analysis in R.

### 2.9. Visualization and Statistical Analysis

Pairwise comparisons to fit a linear model for each DEG by SM, EC users, and NS status were assessed by limma [48] function *lmFit* for the 420 inflammatory genes. Then, the *contrasts.fit* function was applied to results from *lmFit*, which allowed for computation of estimated coefficients and standard errors for pairwise comparisons across the three smoking groups. Empirical Bayes (eBayes) was applied to the results from *contrasts.fit* to calculate log2 fold change and *p*-values using a moderated t-statistics. The results of eBayes analysis were gathered using the function *topTable,* and the *p*-values were adjusted for multiple comparisons using FDR. Genes with FDR adjusted *p*-values < 0.05 and absolute log2 fold change > 1 were considered to be significant differentially expressed inflammatory genes (DEGs). DEGs from pairwise comparisons analysis were then input into QIAGEN Ingenuity Pathway Analysis (IPA). IPA was provided with gene names and log2-fold change difference of our DEGs to analyze canonical pathways and function. Absolute IPA z-scores > 2 were used to consider a pathway as significant.

DEG by SM, EC users, and NS were visualized by heatmaps using log2 normalized counts of DEGs using the pheatmap package v.4.0.4 [49] in R. Gene counts were scaled and plotted based on hierarchical clustering, with annotations based by smoking status. Pearson’s correlations were calculated using the *cor* function on each smoking group’s normalized gene counts. Correlation heatmaps were then plotted using the pheatmap package v.4.0.4 in R, with gene order for each intragroup correlation based on the hierarchical cluster order from an analysis using all three smoking statuses.

### 2.10. Bacterial-Immune Response Association Analysis by IntLIM

Correlation analysis of bacteria genus by macrophage subtype composition and human gene expression was conducted using IntLIM [50]. Associations were considered significant using IntLIM default parameters (difference in correlation (diffcorr) > 0.5 and FDR adjusted *p*-values < 0.1). In the first linear model, IntLIM was provided with bacteria genera counts and macrophage subtype composition proportions from CIBERSORT for bacteria genera-macrophage subtype association analysis. The model for this association analysis was *macrophage_subtype ~ bacteria_genera + smoking_status + bacteria_genera:smoking_status*. A second linear model was used to determine bacteria genera and inflammatory DEG correlations by IntLIM. The model for this association was *gene ~ bacteria_genera + smoking_status + bacteria_genera:smoking_status*. The analysis, however, was not conducted for the EC and NS comparison because there was only 1 DEG.

## 3. Results

### 3.1. CIBERSORT Immune Cell Subtype Analysis

Based upon immune cell subtypes computed by CIBERSORT, we observed a clear separation between SM and NS (Appendix A). Upon further analysis of CIBERSORT’s results, we observed that macrophages composed, on average, 68% of cells. The macrophage subtypes by SM, EC user, and NS groups are shown in Table 2, where the undifferentiated M0 macrophages were significantly increased for SM and EC users compared to NS (FDR < 0.0167), but there was no difference between SM and EC users (Figure 1A). For the pro-inflammatory M1 macrophage subtype, a significant decrease was observed in SM compared to NS, while no significant difference was found for the EC users compared to SM and NS (Figure 1B). For the anti-inflammatory M2 macrophages, we observed a significant decrease in M2 macrophages in both SM and EC users compared to NS (Figure 1C). Amongst the other immune cell types, significant differences were observed between SM and NS in CD4+ T-cells—memory resting, T cells—regulatory, and dendritic cells—resting (Appendix A). Significant differences were also found between SM and EC users for CD4+ T-cells—memory resting and dendritic cells—resting. There were no significant differences for NK cells, B cells, CD8+ T cells, monocytes, mast cells, eosinophils, and neutrophils (data for all 22 CIBERSORT identified subtypes are shown in Appendix A).

### 3.2. Inflammatory Gene Expression Analysis

There were 68 DEGs identified in the SM/NS comparison (FDR < 0.05 and a fold-change of at least 2 (log2 fold change > 1)) (Figure 2A,D; Appendix A). For the SM/EC user comparisons, there were 19 DEGs (Figure 2B,D; Appendix A). Eighteen of these were common to both comparisons, which were *ADORA3*, *AOC3*, *C8B*, *CCL20*, *CHI3L1*, *CXCL10*, *CXCL11*, *CXCL9*, *CYP4F11*, *FCER1A*, *IDO1*, *IRG1*, *LIPA*, *MGLL*, *NCF1*, *PLA2G7*, *SERPING1,* and *VNN1*, representing genes in the chemotaxis, allergy, tissue remodeling, and other pathways. Of these 18 genes, *ADORA3*, *CHI3L1*, *CYP4F11*, *FCER1A*, *LIPA*, *NCF1*, *PLA2G7,* and *VNN1* were observed to have increased expression in SM. The 19th DEG observed in the SM/EC comparison was *FN1* (fibronectin), which was not significant in the SM/NS comparison and also was higher in the EC users than SM. There was only one DEG for the EC/NS comparison that met this criteria (Figure 2C,D; Appendix A), which was *CSF-1*.

In hierarchical clustering analysis, seven of the eight SM clustered together along with two EC users, and nine of the ten NS clustered with only two EC users and one SM, while six of the ten EC users were separated and clustered with one NS (Figure 2E).

A heatmap showing which of the 68 DEG correlation profiles for each DEG to each other by smoking status is shown in Figure 3A–C. The greatest correlative clustering change in the lower right of each figure is outlined in black, representing 30 DEGs, which decreased in intensity from NS to EC to SM. The highly correlated 30 genes were identified as part of canonical pathways, including Th1 and Th2 differentiation pathways. We also observed a second shift in correlation profiles in the upper left quadrants that increased in intensity from NS to EC use to SM. These genes were not identified as part of a particular immune cell canonical pathways, but the genes were identified as part of the atherosclerosis signaling pathways. The shift in gene correlation based on smoking status was also observed in the density plot (Figure 3D). The SMs have a smaller peak for highly correlated genes (Pearson Correlation > 0.5) compared to EC users and NS.

Pathway analysis for the 69 DEGs was conducted using IPA and an absolute z-score > 2 for significance. Only one pathway was returned in the SM/NS comparison, which was the Th2 (T-helper cells type 2) pathway, with a z-score of −2.34. No pathways were returned discovered in the SM/EC comparison.

### 3.3. Association of Bacteria Genus with Macrophage Subtypes

Linear models were used to find associations between bacteria genera and all macrophage subtypes by smoking status groups. We observed two genus-M0 macrophage subtype pairs in SM/NS, one genus-M0 macrophage subtype pair in EC/NS, and zero genus-macrophage subtype pairs in SM/EC. The two genera-macrophage subtype pairs in SM/NS were *Geminocystis* (Figure 4A) and *Salinivirga* (Figure 4B), which were both associated with M0 macrophages, while the one genera-macrophage subtype pair in EC/NS was *Negativicoccus,* which was also associated with M0 macrophages (Figure 4C). For the *Geminocystis*-M0 macrophage association, in NS, an increase in *Geminocystis* was associated with an increase in M0 macrophages, while, in SM, a decrease in *Geminocystis* was associated with an increase in M0 macrophages. In the *Salinivirga*-M0 macrophage association, for both NS and SM, an increase in *Salinivirga* was associated with an increase in M0 macrophages. No association of *Geminocystis* or *Salinivirga* with macrophage subtypes was observed in EC users. For the *Negativicoccus*-M0 macrophage association, NS had a decrease in *Negativicoccus,* and EC users had an increase in association with an increase in M0 macrophages.

### 3.4. Association of Bacteria Genus and Differentially Expressed Inflammatory Genes

Linear models with an interaction term were used to identify smoking status group-specific bacteria genera associations with inflammatory DEGs. This association analysis used the bacteria genera list and the DEGs lists (68 DEGs SM/NS, 19 DEGs SM/EC) from our pairwise comparisons. We observed three genus–gene pairs in SM/NS and zero genus–gene pairs in SM/EC (FDR adjusted *p*-value < 0.1 and correlation difference of >0.5). These three genus–gene pairs were *Candidatus Solibacter*-*GATA3* (Figure 4D), *Moraxella*-*GATA3* (Figure 4E), and *Blastococcus*-*XCR1* (Figure 4F). We observed a decrease in *GATA3* expression for SM and an increase for NS associated with increased *Candidatus Solibacter*. For *Moraxella-GATA3* association, we observed a decrease in *GATA3* expression with increasing *Moraxella* in SM, and no linear trend was observed in NS. *Blastococcus* was positively correlated in SM with *XCR1* expression, but it was negatively correlated in NS.

## 4. Discussion

This pilot study is the first to concurrently assess associations in NS, SM, and EC use for lung inflammation by immune cell subtype and inflammatory gene expression, as well as to further examine associations with the lung microbiome. In a prior publication, we reported that the bacterial diversity was no different for SM, EC, and NS, and while there were some differences in differentially abundant bacteria among the groups, smokers tended to have lower levels compared to EC and NS, and that total macrophages were higher in SM compared to NS [31]. In this report, both SM and EC use, compared to NS, were associated with about a two-fold proportional increase for undifferentiated naïve M0 macrophages and a concordant decrease in anti-inflammatory M2 macrophages. In contrast, inflammatory gene expression for EC use was more closely aligned with NS than SM. We observed that *FN1*, coding for fibronectin, may be associated uniquely with EC use. The correlation profiles of lung inflammatory gene expression indicated clear differences in the lungs of SM users, EC users, and NS users. Further, bacterial metatranscriptomic analysis identified bacterial genera associated with decreased expression of inflammatory genes *GATA3* and increased expression of *XCR1* in SM. Overall, metatranscriptomic analysis did not support a clear role for the microbiome to be associated with changes in inflammation.

While smoking has both pro-inflammatory and anti-inflammatory effects on the lung, our results suggested that smoking has mostly pro-inflammatory effects on the lung. Previously, cigarette smoking has been associated with lung inflammation, modulating the risk of respiratory disease and lung cancer [6,51]. Smoking is associated with increases in lung innate immune cells, including macrophages and neutrophils [3], and the adaptive immune cells CD8+ T-cells [6,52], but also with decreased CD4+ T-cells [52]. Smoking has anti-inflammatory effects that includes reprogramming from pro-inflammatory M1 macrophage to anti-inflammatory M2 macrophages [53,54].

The significant increase in the percentage of M0 cells for SM compared to NS indicated a macrophage developmental arrest, which counterbalanced the decrease in anti-inflammatory M2 macrophages, and while there was also a decrease in the proportion of pro-inflammatory M1 macrophages, the magnitude of the proportional decrease was much less. In other studies, and in the dataset reported herein, smoking increased total macrophages [8,15], down-regulated M1 macrophage states, and induced M2-like macrophage states in cultured human inflammatory cells [8,9,10,54,55,56] and mice [8,57], consistent with an anti-inflammatory effect. Eapen et al. (2017) [8] studied both BAL and tissue, although they only reported M1 and M2 macrophages without employing a technology to study M0 populations. They found an increase in total macrophages for SM, no difference for M1, and an increase in M2. Kollert et al. (2009) [58] reported only levels of M2 macrophages in BAL, noting a decrease in M2 for SM compared to NS, consistent with our data. The only study we are aware of that considered smoking associations with M0 macrophages was reported by Bazzan et al. (2017) [59], who studied lung tissues from SM compared to NS, but not inflammatory cell infiltrates in BAL, and they observed a decrease in M0 and an increase in M1 and M2. All of the above studies used methodologies assessing a limited number of immune cell subtypes. Given that smoking, and nicotine, have both anti-inflammatory and pro-inflammatory effects, the increase in M0 percent and macrophage arrest in BAL, as we have found, could promote an overall pro-inflammatory microenvironment. Macrophage subtypes differ in the expression of chemokine receptors and response to chemokines [60]. Whether a difference in chemokines is related to smoking may be one explanation for the results reported herein. Either way, concordant with these findings was an increase in Tregs and dendritic cells, but a decrease in CD4+ lymphocytes. These results indicated that EC users immune cell compositions were mostly similar to SM. Eight of ten EC users were former smokers, which combined with the immune cell composition results of EC users being similar to SM, which may reflect a residual smoking effect.

While the CIBERSORT cell type data indicated that EC use and SM had similar immune cell composition associations, and were different compared to NS, a different pattern of inflammatory gene expression was found. Correlation analysis indicated unique inflammation profiles for SM users, EC users, and NS users. There were 68 significant DEGs, which were inflammation-related genes pathways for SM/NS, of which 18 DEG overlapped with SM/EC comparisons, possibly indicating a residual smoking effect for these. Among the 18 DEGs, the following are decreased in SM: *CXCL9*, *CXCL10,* and *CXCL11*. Meanwhile, DEGs, such as *CHI3L1* and *ADORA3,* are increased in SM. *CXCL9* is expressed by M1 tumor-associated macrophages and recruited CD8+ tissue-resident memory T cells, and increases in both immune cell types in tumors are associated with better overall lung cancer survival [61]. *CXCL9*, *CXCL10,* and *CXCL11* are upregulated in macrophages following immune checkpoint inhibitor treatment and associated with increased CD8+ T-cell infiltration into tumors [62], and they are found to be downregulated in COPD macrophages [54]. High serum levels of *CHI3L1* are associated with poorer prognosis and decreased overall survival for various cancers [63], including lung cancer [64,65]. Given *CHI3L1*’s role in remodeling after injury, increased *CHI3L1* serum levels are also associated with metastasis in non-small cell lung cancer [66]. Another remodeling gene, *ADORA3,* was found to have an expression that was increased in COPD subjects [54]. The single overlapping DEG for SM/NS and EC/NS comparisons was *CSF-1* (macrophage colony stimulating factor 1). This gene codes for an extracellular cytokine, serving as a hematopoietic growth factor affecting the production, differentiation, and function of macrophages [67]. Higher levels of *CSF-1* in SM and EC users suggests that there is a shared pathway in both atherosclerotic and cancer pathways, potentially fostering tumor growth [68]. There was only one gene that was uniquely differentially expressed between SM and EC users, which was *FN1*, which codes for fibronectin, and which was higher in EC users. Fibronectins have a role in maintaining cell structure, maintaining the extracellular matrix, and binding cell surfaces [69,70,71,72]. Conflicting studies have found that fibronectin expression was both down- and up-regulated in mice exposed to e-aerosols [73,74]. Whether EC use has a unique effect on fibronectin expression, which is unrelated to SM, requires further study.

IPA analysis identified an inhibition of the Th2 pathway in SM (Th2 cells as a subtype are not reported in CIBERSORT). Th2 cells are a subset of CD4+ cells, and we found that CD4+ memory resting cells were decreased in SM compared to the EC users and NS, and the Tregs were increased. Th2 cells are responsible more for humoral immunity than for inflammation [75], and the cytokines they secrete (IL-4, IL-5, IL-9, IL-10, and IL-13 [75,76]) were not found to be different in the lungs of SM users, EC users, and NS users [15]. These cytokines also were not affected in NS, using EC in a randomized clinical trial, and using EC [14]. Thus, the significance of the association to the Th2 pathway is unclear. Additionally, noted in the analysis was a statistical increase in resting dendritic cells for SM compared to the EC users and NS, with about a two-fold increase. These antigen-presenting cells may be increased due to the higher levels of cigarette smoke toxicants compared to e-aerosols [77].

Herein, we examined microbiome correlations with CIBERSORT immune cell data by SM, EC, and NS status. Only two genera-M0 subtype correlations were observed for SM/NS, namely, *Geminocystis* and *Salinivirga,* neither of which are considered pathogenic. The *Geminocystis*-M0 correlations were in different directions with a positive correlation in NS and an inverse correlation in SM. The *Salinivirga*-M0 macrophage correlation was positive for both SM and NS. There were no M0 correlations for *Geminocystis* or *Salinivirga* with EC use. For EC/NS, there was a *Negativicoccus*-M0 macrophage inverse correlation for NS and a positive correlation for EC use. *Negativicoccus* has been found in normal lung and in lung carcinomas [78]. These findings are for EC use only, and they may reflect a unique EC effect, including decreased macrophage phagocytosis, as has been observed in vitro for nontypeable *Haemophilus influenzae* [19] and *Mycobacterium tuberculosis* [79].

In this study, the metatranscriptomic data were found to significantly correlate with only three bacteria genera and DEG in SM, and there were no correlations in EC users. Among these, the *Moraxella* genus, which includes *Moraxella catarrhalis,* which is a well-documented lung pathogen associated with defective macrophage phagocytosis [80], was correlated with decreased *GATA3* expression in SM, but not in NS. *GATA3* is a highly bioactive transcription factor involved in both inflammatory and humoral immunity, and it has an important role in differentiation of Th2 effector cells [81]. Thus, as with the Th2 pathway analysis, smoking may affect the lung microbiome via an impact on *GATA3*. For the other correlations, there is no known pathogenicity of *Blastococcus* and *Candidatus Solibacter* in humans.

Although there were several novel findings in this study, some limitations to data interpretation are notable. This study was necessarily small due to the intensive study based on RNA-seq, and so generalizability to SM and EC users, including persons age > 30, is limited. Due to the small sample size, there were also limits on statistical power, as the sample size is insufficiently powered to detect small differences, as well as precluding analyses in SM and EC users by magnitude of use (dose–response effect). By excluding older SM and EC users, we also excluded consideration of concurrent lung damage from long term use. However, the participants we studied here are the demographic which most commonly uses EC. There also was an imbalance by gender in the NS, although it is unknown if the lung microbiome would differ by gender. Another limitation of this study was that CIBERSORT immune cell population analysis is only able to estimate cell population proportions, and thus we did not have absolute cell population data, and our prior studies have shown that total macrophage levels increase with smoking, and less so with EC use, compared to NS [15]. Future studies are warranted to isolate immune cells and perform functional and protein-based assays to understand the impact of SM and EC on the airway inflammatory process. This study method by BAL does not distinguish the bacterial communities that may be different in the alveoli and bronchi. Finally, while our data are consistent with other published data [16,17,18], and in a small pilot trial we showed that EC use in NS increases inflammation [14], the cause and effect relationship of smoking or EC use with changes in inflammation needs to be confirmed in studies of SM switching to EC or quitting in randomized trials.

## 5. Conclusions

In summary, in this cross-sectional pilot study using bronchoscopy biomarkers of inflammation, we observed that SM and EC use was associated with increased M0 macrophage arrest, as well as an almost equal decrease in anti-inflammatory macrophage development. In contrast, SMs were distinct from EC users and NSs for inflammatory gene expression. Inflammation is an important pathway for the development of chronic respiratory disease and lung cancer. The SM and EC associations with inflammation were not explained by alterations in the lung microbiome. The preliminary data support the hypothesis that smoking and EC use have toxic lung effects, influencing inflammatory responses, but not via changes in the microbiome. The importance of macrophage subtype differences on lung cancer risk and other diseases warrants further exploration.

## Figures and Tables

**Figure 1 microorganisms-11-01405-f001:**
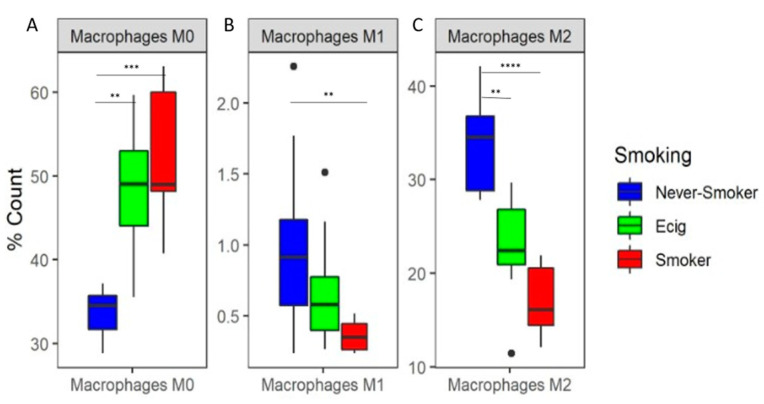
Predicted Percent Composition of Macrophage Subtypes in BAL. Boxplots of predicted percent composition by CIBERSORT of (**A**) M0, (**B**) M1, and (**C**) M2 macrophage subtype by smoking groups. Asterisks represent significance between pairwise smoking group comparison (** *q*-value < 0.01, *** *q*-value < 0.001, **** *q*-value < 0.0001 Dunn test).

**Figure 2 microorganisms-11-01405-f002:**
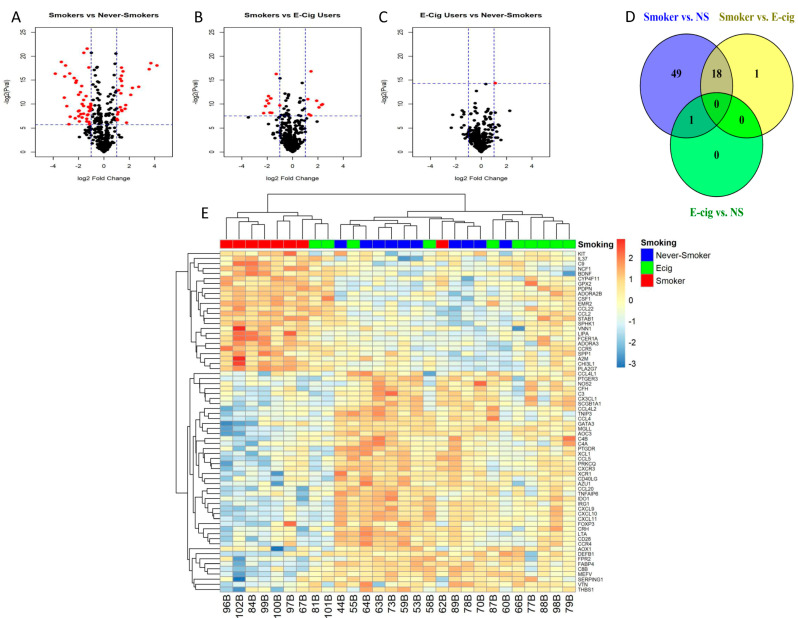
Inflammatory gene expression patterns in response to smoking or e-cig use. (**A**–**C**) Volcano plots depict log2 fold change vs. –log2(*p*-value) for differentially expressed genes (DEG) for each smoking status pairwise comparison (red is statistically significant and black is not significant), (**A**) smokers vs. never-smokers, (**B**) smokers vs. e-cig users, and (**C**) e-cig users vs. never-smokers. Significant DEGs are colored red (horizontal dashed lines represent raw *p*-values that correlate to FDR < 0.05, and vertical dashed lines represent absolute log2 fold change > 1). (**D**) Venn Diagram of # of DEGs by smoking status pairwise comparison. An amount of 68 DEGs were observed in the smoker vs. never-smoker comparison, and 19 DEGs were observed in smoker vs. e-cig user comparison, and one DEG is observed in the e-cig user vs. never-smoker comparison. There are 18 genes that are in common between the comparisons of smoker vs. never-smokers and smokers vs. e-cig users. There is one gene in common between the comparison of smoker vs. never-smokers and e-cig users vs. never-smokers. (**E**) Heatmap of 68 DEGs in all study subjects. Seven of eight smokers cluster together with two e-cig users, and the cluster in the middle contains nine of the ten never-smokers, with two e-cig users and one smoker, and the cluster on the right contains six of ten e-cig users and one never-smoker.

**Figure 3 microorganisms-11-01405-f003:**
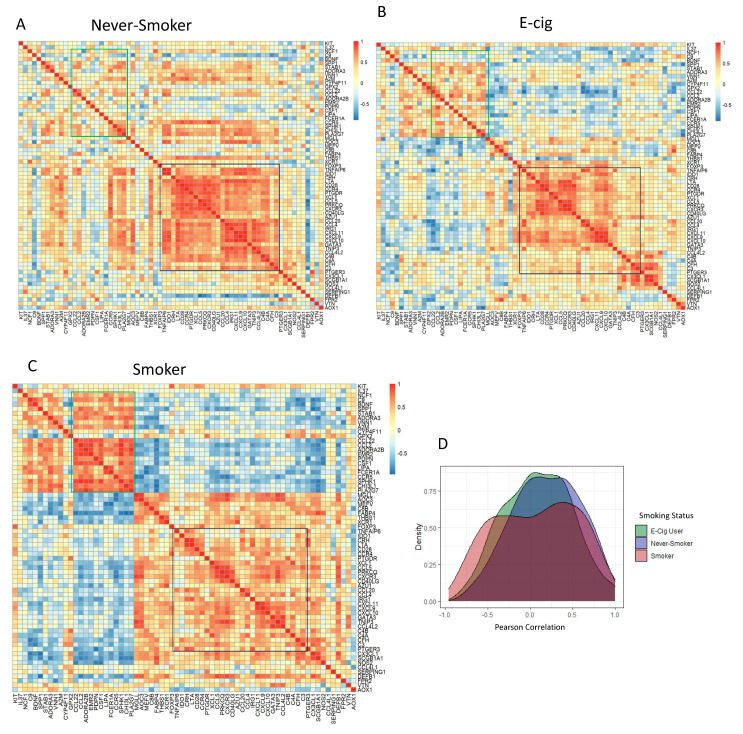
Shift in Gene Expression Observed by Pearson Correlation Heatmaps and Density plot of DEGs by Intragroup Comparison. Correlation heatmaps of 68 DEGs in (**A**) never-smokers, (**B**) e-cig users, and (**C**) smokers. Highly positive correlations in genes are represented by shades of red, while highly negative correlations in genes are represented by shades of blue. The dark blue outlined region located on the right side of all correlation heatmaps represents genes that shifted from highly correlated in never-smokers to less correlated in smokers. The green outlined region located in upper left of all correlation heatmaps represents genes that were lowly correlated in never-smokers that became highly correlated in smokers. (**D**) Density plot representing Pearson correlation vs. density of all three intragroup correlations. Compared to never-smokers and e-cig users, a decrease in high correlations (Pearson Correlation > 0.5) was observed in smokers.

**Figure 4 microorganisms-11-01405-f004:**
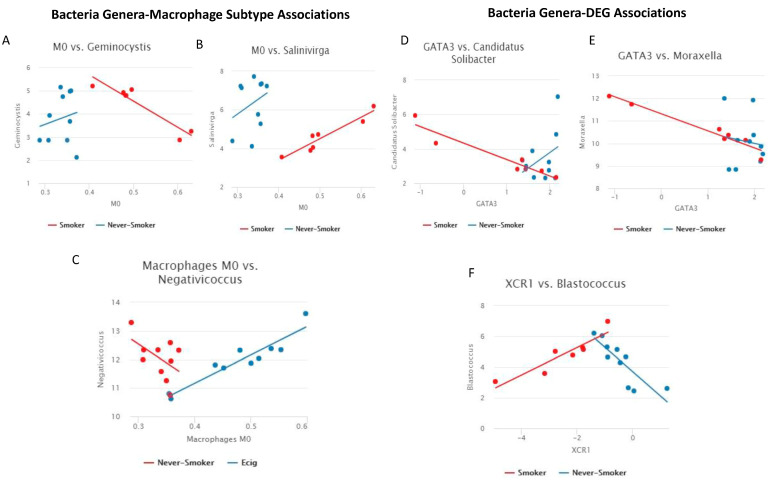
Line plots of bacteria genera-immune response associations. (**A**–**C**) Line plot of macrophage subtype vs. bacteria genus (Significant pairs were calculated as difference of Pearson’s correlation > 0.5 and FDR *p*-value < 0.2). Comparison is SM/NS for A and B, comparisons is EC/NS for C. (**A**) Geminocystis-M0 macrophages: a negative linear trend was observed in SM with positive linear trend observed in NS. (**B**) Salinivirga-M0 macrophages: a positive linear trend was observed in SM and NS. (**C**) Negativicoccus-M0 macrophages: a negative linear trend was observed in NS, and a positive linear trend was observed in EC users. (**D**–**F**) Line plot of gene expression vs. bacteria genera (Significant pairs were calculated as difference of Pearson’s correlation > 0.5 and FDR *p*-value < 0.2). All comparisons are between SM/NS. (**D**) Candidatus Solibacter-GATA3: a negative linear trend was observed in SM, and a positive linear trend was observed in NS. (**E**) Moraxella-GATA3: a negative linear trend was observed in SM, with no linear trend observed in NS. (**F**) Blastococcus-XCR1: a positive linear trend was observed in SM, and a negative linear trend was observed in NS.

**Table 1 microorganisms-11-01405-t001:** Clinical Characteristics of Study Participants.

	NS (*n* = 10)	EC (*n* = 10)	SM (*n* = 8)	*p*-Value (Chi-Square)
Age (range)	21–30	21–29	21–30	0.22
Mean (±SD)	25.6 (2.8)	27.5 (1.9)	25.9 (2.7)	
Gender				0.38
Male (%)	6 (60%)	6 (60%)	7 (87.5%)	
Female (%)	4 (40%)	4 (40%)	1 (12.5%)	
Race				0.41
White (%)	8 (80%)	8 (80%)	8 (100%)	
African American/Asian (%)	2 (20%)	2 (20%)	0 (0%)	
Bronchoscopy Site				0.84
Left Lung (%)	5 (50%)	5 (50%)	5 (62.5%)	
Right Lung (%)	5 (50%)	5 (50%)	3 (37.5%)	
Library Generation Batch				0.89
First Batch (%)	6 (60%)	6 (60%)	4 (50%)	
Second Batch (%)	4 (40%)	4 (40%)	4 (50%)	

**Table 2 microorganisms-11-01405-t002:** Macrophage Subtypes by Smoking Status. Median and range for M0, M1, and M2 macrophages subtypes in never-smokers, e-cig users, and smokers. Overall *p*-values were calculated by the Kruskal-Wallis and were FDR corrected.

CIBERSORT (%) Cell Counts					
	Never-Smoker (*n* = 10)	E-Cig User (*n* = 10)	Smoker (*n* = 8)		
**Cell Type (%)**	Median (Range)	Median (Range)	Median (Range)	**Raw *p*-Value**	**FDR-Corrected** ** *p* ** **-Values**
**Macrophages M0**	34.5 (28.8–37.1)	49 (35.5–59.7)	49 (40.7–63.1)	**2.20 × 10^−4^**	**2.09 × 10^−3^**
**Macrophages M1**	0.91 (0.24–2.26)	0.58 (0.26–1.51)	0.35 (0.23–0.51)	**7.56 × 10^−3^**	**2.40 × 10^−2^**
**Macrophages M2**	34.6 (27.8–42.1)	22.4 (11.4–29.7)	16.1 (12.1–21.9)	**8.42 × 10^−5^**	**1.60 × 10^−3^**

## Data Availability

Data are available through GEO—Gene Expression Omnibus GSE227547.

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
