# Peer review of "A Pilot Cross-Sectional Study of Immunological and Microbiome Profiling Reveals Distinct Inflammatory Profiles for Smokers, Electronic Cigarette Users, and Never-Smokers"

_microorganisms, 2023, doi:10.3390/microorganisms11061405_

Round 1
Reviewer 1 Report
To the Editor,
In their interesting study Shields PG and Mathé EA, et al. used RNA-sequencing of bronchoalveolar lavage fluid (BALF) cells sampled from healthy non-smokers (NS, n=10), current smokers (SM, n=10), and e-cigarette users (EC, n=8), to investigate lower airway inflammatory and microbiome patterns associated with smoking and vaping. This study is a continuation of an earlier report from the same group describing bacterial diversity in lung and oral microbiomes. Here, they also compare mRNA transcriptome to investigate differentially expressed genes (DEGs) and assess fractions of various cell types in BALF. The Authors used deconvolution of bulk RNA-seq data (CIBERSORT) to identify subtypes of macrophages. Although this is an indirect method, in my experience, the expression of cell signature mRNA markers correlates strongly with the respective cell counts in BALF. Anti-inflammatory M2 macrophages decreased considerably both in smokers and users of electronic cigarettes compared to NS. This suggests a predominantly pro-inflammatory pattern associated with smoking, similar to that induced with e-cigarettes. However, gene expression profiles (DEGs) were more diverse with several chemokine genes involved in inflammation (eg, CXCL10) rather downregulated in BALF cells, mostly in SM. Finally, the Authors attempted to link microbiome profiles with macrophage subtypes and DEGs identified in group comparisons. This revealed quite similar associations, with only a few bacteria genera showing different correlations depending on SM/EC/NS status. Therefore, the Authors reject the hypothesis that inflammatory profiles related to smoking are mediated by alterations in the lung microbiome. This study is well designed and conducted, with an extensive Methods section providing the necessary details. The conclusions are based on the results obtained. The apparent similarity in lower airway inflammation patterns in combustible cigarette and e-cigarette users is the key result with potential clinical implications. It may be of interest to researchers dealing with effects of e-cigarette aerosol exposure. I don’t have many critical comments. Below, I list just a few suggestions to be considered by the Authors and Editor.
Minor comments
1. As shown in the supplement, there were other changes in the predicted cell composition. Why not start with a general overview of the BALF cellular profile, including predicted abundances of each cell type and possibly the principal component or, preferably, the tSNE visualization. This should reveal considerable differences in macrophage subsets and might explain why the Authors focused primarily on macrophages. Such cellular surveys of BALF samples based on seq data were not frequently published and are much needed.
2. The Authors found considerable changes in the percentage of BALF macrophage subsets in the e-cigarette and smokers groups. In particular, there was an increase in the predicted count of M0 macrophages and a decrease in M2 macrophages, compared to non-smokers. As the predicted counts are actually based on the abundance of mRNA signatures of each subtype, it should be expected that key mRNA markers will also show a similar trend. For example, IDO1 (based on the literature it predominates in M2) was ~4-fold decreased in SM compared to NS, which corresponds to a nearly 3-fold decrease in M2 count. The authors could update Fig. 1 with expression data (eg, normalized read counts or qPCR validation) for crucial markers of M0, M1 and M2. This will help link the predicted macrophage subtypes with the respective markers and in some instances may complement the DEG data shown in the next chapter.
3. The Authors suggest that changes in macrophage subsets arise likely to changes in differentiation (developmental arrest, line 363). Macrophage subtypes differ in the expression of chemokine receptors and response to chemokines (eg, PMID: 25359998). Several chemokines and receptors were differentially expressed in SM compared to other groups. Could this pattern reflect differences in macrophage recruitment and thus explain some of the observed changes in subsets?
4. The Authors should consider adding gene labels for top DEGs in V-plots (Fig 2A-C) and for key genes in the heat-map (Fig. 2E). Additionally, Fig 2D is a mixture of different DEGs; should it be rather split to show separately up- or down-genes?
5. The approach to analysis shown in Results 3.3 is difficult to understand for readers not familiar with IntLIM packages. I suggest adding some short accessible comment in this section to explain the idea behind this analysis. For example, what does it mean that they observed ‘one genus-M0 macrophage subtype pair in EC/NS’ (line 301)? Based on the Methods section (line 212), they first compared/correlated ‘bacteria genera [normalized read?] counts and macrophage subtype composition proportions’. In such a way, they could initially provide the results of the correlation analysis for the entire cohort (top r-coeff., general associations of the microbiome and M-subsets), and possibly separately for the subgroups NS, SM, and EC (to highlight differences). What is shown currently (if I am correct) is just listing the few genera that showed the highest difference in r-coeffs (r-diff >0.5, significant at FDR <0.2) obtained in certain two groups. But the broader context is lacking, I mean, just showing also the similarities with hundreds of genera not correlated or similarly correlated with M0 in all studied groups. Could the Authors provide some summary plots here. For example, (1) comparing r-coeffs (M0 vs genera) in the given two groups (eg, NS vs. SM), or (2) showing ‘difference in r-coeffs’ vs. P-value with the few identified (most different correlations) bacteria genera labeled. Again, this is only my suggestion.
6. When discussing DEG data, the Authors should mention which genes were up or downregulated, eg, in SM (line 248). For example, in line 393, important DEGs (out of 18) linked to SM are listed, of which CXCL9, CXCL10, CXCL11 were decreased in BALF cells from SM, while CHI3L1 was strongly increased (~7-fold). Considering the latter, airway chitinase like proteins were previously linked with active smoking and COPD (particularly in patients with marked airway inflammation).
7. GATA3 has been regarded as a regulator of M2 polarization (eg, PMID:27354564, PMID:31935430). Please, discuss this in the context of microbiome-GATA3 associations detected in smokers.
8. Please, update Table 1 with BALF cell differential data, including absolute cell counts. Actually, an increase in total macrophages is pointed in the discussion (line 367), but data not shown. This is only a suggestion, but not a requirement to present these data. I am aware that the Authors might not have access to a sufficient sample size to do both conventional cytospin or cytometry staining and RNA extraction.
9. The previous study by this research group on the bacterial diversity (PMID:35667088) is not included in the references.
10. Abstract line 23, please add ‘in the bronchoalveolar lavage cells’. Currently, it is not specified in the abstract what kind of sample obtained by bronchoscopy was studied.
11. Lines 490-491, saliva was not analyzed in this manuscript.
Reviewer 2 Report
The manuscript “A Pilot Study of Immunological and Microbiome Profiling Reveals Distinct Inflammatory Profiles for Smokers and Electronic Cigarette Users” was submitted to Microorganisms.
The authors deal with an interesting topic; however, some adjustments must be considered.
Title: Please include that it is a cross-sectional study. Furthermore, considering the limitations of the study, the authors should be less enthusiastic about the proposed title.
Abstract
Objective: You must set a clear objective. It seems that an a priori association between SM and EC is indicated.
Methods: Please define M0 (undifferentiated) and M2 (anti-inflammatory).
Results: lines 28, 29. “correlated positively and inversely with M0 and M2 macrophages respectively”. Please present the results of this correlation (r=), with their respective level of significance.
Conclusions: It should be noted that the results are based on the limitations of the study and should be viewed with caution.
Keywords: metatranscriptome is not a MeSH term.
Introduction
Objective: It seems that an a priori association between SM and EC is indicated.
A clear objective must be proposed.
Please always define NE in the same way. Sometimes it is described as a never-smoker and sometimes as a non-smoker.
In addition to the reference number, in several places, you also include the authors. Please correct this throughout the manuscript.
Methods
Line 79. Please define BAL.
Line 82. Present the randomization process in detail.
Line 150. The use of non-parametric (K-W) and later parametric (Pearson's correlation) tests is indicated. Before that, the statistical test used to establish the normal distribution of the data with its respective confidence interval must be presented.
Line 153. Please indicate the reasons for using this cut-off point (p<0.0167).
Results
Figure 3. Lines 286-287; 288-289; 290-291; 292-293. Please present the results of these correlations (r=), with their respective level of significance.
Conclusions
It should be noted that the results are based on the limitations of the study and should be viewed with caution.
Lines 469-470. “In summary, in this cross-sectional pilot study using bronchoscopy biomarkers of inflammation, we observed that”. This information is unnecessary in this section.
Author Response
Please see attached - correct file is reviewer 2.doc

Reviewer 3 Report
The authors investigated the associations for smokers (SM) and electronic cigarette (EC ) lung microbiomes in correlation with immune cell subtypes and inflammatory gene expression in 28 individuals having bronchoscopy . Analyses were performed by RNASequencing methodologies . Yet, the CIBERSORT computational algorithm were applied to determine immune cell subtypes, inflammatory gene expression and microbiome metatranscriptomics. SM and EC seems to be associated with increase in undifferentiated M0 macrophages. Smokers microbiome differed from EC users and NS for inflammatory gene expression. Their results showed that SM and EC have toxic lung effects influencing inflammatory impact , but not via shifts in the microbiome.
It is a scientifically sound paper well written based on an extended bibliography.their results are nicely presented in tables and figures summarizing the research .they used advanced methodological techniques for their research and their results were investigated by a computational algorithm.
the paper should be of high interest to the scientific community
my suggestion is to ACCEPT and publish this paper
Author Response
There were no requested changes by this reviewer who wrote: ". . . my suggestion is to ACCEPT and publish this paper"!
Round 2
Reviewer 2 Report
Line 367. Please revise.
Author Response
This astute reviewer noted an error. The manuscript should have said “. . . and that total macrophages were higher in SM compared to NS” rather than “. . .and that total macrophages were higher in SM compared to SM” This is now corrected.